# Epidemiological Characteristics and Spatiotemporal Clustering of Symptomatic Hepatitis E Virus Reinfection in Zhejiang Province, 2005–2023

**DOI:** 10.3390/v16111676

**Published:** 2024-10-26

**Authors:** Lu Zhou, Yijuan Chen, Fengge Wang, Zixiang Chen, Yihan Lu, Ziping Miao

**Affiliations:** 1Department of Epidemiology, Ministry of Education Key Laboratory of Public Health Safety, School of Public Health, Fudan University, Shanghai 200032, China; luzhou22@m.fudan.edu.cn (L.Z.); fgwang22@m.fudan.edu.cn (F.W.); zixiangchen23@m.fudan.edu.cn (Z.C.); 2Department of Communicable Diseases Control and Prevention, Zhejiang Key Lab of Vaccine, Infectious Disease Prevention and Control, Zhejiang Provincial Center for Disease Control and Prevention, Hangzhou 310051, China; yjchen@cdc.zj.cn; 3Shanghai Institute of Infectious Disease and Biosecurity, Fudan University, Shanghai 200032, China

**Keywords:** HEV reinfection, epidemiological characteristics, spatial autocorrelation, spatiotemporal clustering, Zhejiang Province

## Abstract

Hepatitis E virus (HEV) reinfection is prevalent among the population, posing a significant burden on prevention and control efforts. In this study, we conducted a comprehensive analysis of data from China’s Disease Prevention and Control Information System’s infectious disease surveillance system to identify the epidemiological characteristics, spatiotemporal clustering, and high-risk populations of HEV reinfection. From 2005 to 2023, HEV reinfection in Zhejiang Province exhibited a fluctuating trend, peaking in 2020, with a 3–5-year lag compared to the pattern of HEV incidence. The Cox model indicated that individuals aged 40–50 and females are at higher risk of reinfection. Spatial autocorrelation was observed in reinfection cases from 2011 to 2016, with high–high clustering areas concentrated in downtown Hangzhou. Additionally, spatiotemporal scanning revealed that the clustering of reinfection cases has shifted from Hangzhou to coastal areas in recent years. Our findings suggest that targeted prevention and control measures for HEV rein fection should be implemented based on the characteristics of high-risk populations and spatiotemporal clustering patterns.

## 1. Introduction

Hepatitis E (HE), caused by the Hepatitis E virus (HEV), is an acute, self-limiting infectious disease. It is responsible for approximately 60,000 deaths globally each year and is one of the leading causes of acute viral Hepatitis worldwide, resulting in significant economic and social burdens [1]. In recent years, the incidence of HE has shown an overall upward trend in China, with high-prevalence areas primarily concentrated in the eastern Yangtze River Delta region, making it a public health issue that cannot be ignored [2,3].

A large-scale meta-analysis, which included studies on the seroepidemiology of HE in the Chinese population from 1997 to 2022, indicated that the prevalence of HE in China was high, with a seroprevalence of anti-HEV IgG of approximately 23.17% (95% CI: 20.23–26.25) [4,5]. A seroepidemiological survey conducted in Zhejiang Province showed that the seropositivity rate of anti-HEV IgG in the general population was 55.40%, with a standardized positivity rate of 45.54%, which was higher than the national average mentioned above [6]. Although the world’s first Hepatitis E vaccine, Hecolin (HEV 239 vaccine), was licensed in China and launched in 2012, the vaccination coverage rate remained low [7], resulting in HEV antibody levels in various populations being primarily acquired through natural infection.

The protection acquired from prior HEV infection gradually diminished, which led to widespread HEV reinfection among populations, posing significant challenges to the prevention and control of HE [8]. However, current research on HEV reinfection primarily focused on animal model studies, with no specific studies on human reinfection reported to date. Choi et al. demonstrated the possibility of HEV reinfection using a rhesus monkey model [9]. A large cohort study conducted in rural eastern China found that 17% of symptomatic HE cases were due to reinfection, with the majority of these cases occurring in women and younger individuals [10,11]. However, this study did not specifically investigate the epidemiological characteristics of this population. Seriwatana et al. tested nearly 200 HE cases from countries with endemic HE and found that the IgM-to-IgG ratio in patients with reinfection differed from that in patients with primary infections. The study also noted that asymptomatic reinfection events were more common than symptomatic ones [12]. Given that only 5–30% of HE patients exhibited symptomatic infection [13], it is suggested that widespread reinfection might have occurred among many asymptomatic HE patients, significantly increasing the infection risk for high-risk groups such as pregnant women and the elderly. The risk of HEV reinfection in Zhejiang Province has been underestimated, yet research on HEV reinfection is still deficient. Therefore, this study aimed to identify susceptible areas and high-risk groups by examining the epidemiological characteristics and spatiotemporal distribution patterns of HEV reinfection to provide a basis for implementing more specific preventive measures. Additionally, this work offered evidence-based recommendations to local health departments for optimal allocation of limited resources, enhanced targeted monitoring, and alleviated the disease burden.

## 2. Materials and Methods

### 2.1. Data Sources and Data Collection

This study used the Hepatitis E case database from Zhejiang Province within the Chinese Disease Surveillance Information Reporting System, collecting data on HE cases in the province from 1 January 2005 to 31 December 2023. All cases in this database were notified by experienced medical professionals from various hospitals across Zhejiang Province as per the “WS301-2008 Diagnostic Criteria for Hepatitis E” by China’s National Health Commission. Patients were diagnosed comprehensively based on their epidemiological history, symptoms and signs, and laboratory tests, among which testing positive for anti-HEV IgM is a necessary condition for confirming HEV infection. In the criteria, HEV RNA examination was not included yet. The main information in the database included gender, age, occupation, residential address, onset date, and diagnosis date. Demographic data, such as the resident population of each city, were obtained from the Zhejiang Provincial Statistical Yearbooks from 2005 to 2023 [14].

In this study, patients with reinfection were defined as those who had two or more reported cases of HE infection between 2005 and 2023, with an interval of at least six months between consecutive reports. To avoid misclassification of chronic infection patients as reinfection cases due to multiple diagnoses, we excluded patients with immunodeficiency based on their medical records [15]. Following the inclusion and exclusion criteria mentioned above, two researchers independently performed a double-matching process within the HE case database to identify a list of reinfected individuals in Zhejiang Province from 2005 to 2023. It should be noted that the database included only patients who presented symptoms and/or sought medical attention. Therefore, the term “patients with reinfection” used in this manuscript specifically referred to those who experienced symptomatic reinfection. For consistency and simplicity, we defined them as “patients with reinfection” throughout the text.

### 2.2. Sample Size

Based on previous research and preliminary investigations, we hypothesized that 17% of patients with symptomatic infection were reinfected cases [10]. With a margin of error set at 1%, the calculated minimum sample size required was 5420. Therefore, the over 40,000 samples collected in this study’s database were sufficient to provide adequate statistical power for the analysis.

### 2.3. Statistical Analysis

Descriptive statistics for continuous and categorical data were conducted using means, variances, quartiles, and percentages. One-way ANOVAs or *t*-tests were used for continuous variables, while the χ^2^ test or Fisher’s exact test was employed to assess significant differences in categorical variables. The Cox proportional hazards model was used to evaluate factors associated with HEV reinfection, including demographic covariates such as age and gender. Additionally, epidemiological characteristics were separately extracted for different cities to validate the robustness of the results and perform sensitivity analysis. All analyses were conducted using SPSS Statistics version 25.0 (IBM Corp. Released 2017. IBM SPSS Statistics for Windows, Version 25.0. Armonk, NY, USA: IBM Corp.) or R 4.1.3 (R Core Team (2022). R: A language and environment for statistical computing. R Foundation for Statistical Computing, Vienna, Austria. URL https://www.R-project.org/, accessed on 23 October 2024). A two-sided *p*-value < 0.05 was considered statistically significant.

### 2.4. Spatial Autocorrelation Analysis

ArcGIS Desktop (version 10.7; ESRI, Redlands, CA, USA) was used to apply Moran’s I index to describe the degree of global and local spatial autocorrelation. Moran’s I ranges from −1 to 1; when I > 0, it indicates a positive spatial correlation of HEV reinfection cases, while I < 0 indicates a negative spatial correlation. The I value of 0 suggested a random distribution of cases with no spatial autocorrelation. A two-sided *p*-value < 0.05 was considered statistically significant.

### 2.5. Spatiotemporal Scan Analysis

SaTScan 10.1.2 (Kulldorff M. (1997). SaTScan—software for the spatial and space–time scan statistics. Version 10.1.2. Available from: https://www.satscan.org/, accessed on 23 October 2024) was employed to perform a spatiotemporal scan of HEV reinfection cases in Zhejiang Province from 2005 to 2023 using a discrete Poisson model. The basic time unit for the scan was set to months, with the scanning period spanning from January 2005 to December 2023. The maximum temporal scanning window was set to 50% of the total study duration. The basic geographic unit was set to the city level, with the maximum spatial scanning area set to 20% of the total population. After the scan, the log-likelihood ratio (LLR) value of the windows was tested, and a *p*-value < 0.05 was considered indicative of spatiotemporal clustering.

## 3. Results

### 3.1. Epidemiological Characteristics

A total of 41187 HE cases from Zhejiang Province between 2005 and 2023 were included in the study, among which 877 cases of reinfection were identified, involving 429 individuals. The reinfection rate among symptomatic HE cases was 2.13%. The incidence rate of HEV reinfection (per million) was calculated based on the year-end resident population data from the Zhejiang Provincial Statistical Yearbooks. The annual average incidence rate of HEV reinfection in Zhejiang Province from 2005 to 2023 was 0.813 per million. Apart from 2020, when the number of HE cases decreased due to the COVID-19 pandemic, the incidence of HE has shown a rapid upward trend in recent years (Figure 1). The reinfection rate has gradually increased since 2005, peaking in 2020 before experiencing a sharp decline.

Among the 429 patients with reinfection, 413 experienced two episodes of HE infection between 2005 and 2023, 14 had three infections, 1 had four infections, and 1 had five infections, with all infection intervals being at least 6 months apart, as shown in Table 1. Of the patients with reinfection, 264 were male (61.5%) and 165 were female (38.5%). Compared to the single-infection patients between 2005 and 2023, which included 26145 males (67.2%) and 12765 females (32.8%), there was a significant difference (χ^2^ = 6.15, *p* = 0.013), with a notably higher proportion of females among the patients with reinfection. Regarding age distribution, the median age at initial infection for patients with reinfection was 46.0 (37.0, 56.0), which was significantly younger than the 50.0 (39.0, 61.0) for single-infection patients (F = 12.516, *p* < 0.001). In terms of age group distribution at initial infection (Figure 2), the proportion of patients with reinfection aged 41–50 years (n = 130, 30.3%) was significantly higher compared to single-infection patients (n = 8649, 22.2%). Conversely, the proportion of patients with reinfection aged 61–70 years (n = 37, 8.6%) was significantly lower than that of single-infection patients (n = 5840, 15.0%). Among the patients with reinfection, those with three infections had a higher proportion of individuals aged 11–20 years (n = 2, 14.3%) and a lower proportion of individuals aged 41–50 years (n = 2, 14.3%) compared to those with only two infections. Statistical comparisons of the ages at the first two infections and the age at the third infection for patients with three infections revealed that these patients were generally younger overall (Figure 3). In terms of occupation distribution, the largest proportions among patients with reinfection were farmers (n = 150, 35.0%) and commercial service (n = 58, 13.5%). In contrast, single-infection patients had the largest proportions as farmers (n = 17,551, 45.1%) and workers (n = 4725, 12.1%). The proportion of farmers among patients with reinfection was significantly lower than that among single-infection patients, while the proportions of staff, food industry, and commercial service were significantly higher (χ^2^ = 64.923, *p* < 0.001). Compared to patients with two infections, the proportion of farmers among those with three infections decreased, while the proportion of retired personnel increased.

Sensitivity analysis of the demographic characteristics of patients with reinfection in Hangzhou, Wenzhou, and Taizhou (Appendix A) showed consistency with the above results, indicating good stability.

From a geographical perspective (Figure 4), HEV reinfection cases in Zhejiang Province were primarily concentrated in Hangzhou, Taizhou, and Wenzhou, with fewer cases in regions like Lishui and Jiaxing. Among these, Hangzhou had a significantly higher number of cases compared to other cities, and the proportion of reinfection cases in Wenzhou and Taizhou was notably higher than that of single-infection cases. When analyzing the cases by the year of onset (Figure 5), it was found that the number of reinfection cases peaked around 2018 and has decreased in recent years. We calculated the annual number of HEV infection cases in all cities in Zhejiang Province (Appendix A). The study found that the number of initial infections and reinfections (second and third infections) per 1,000,000 people in Hangzhou was significantly higher than in other cities. Additionally, compared to other cities, Taizhou had higher rates of initial infections, while Taizhou and Wenzhou had higher rates of reinfections in recent years.

In addition to the basic characteristics, we also analyzed the infection intervals (months) for cases with different initial onset years, including multiple intervals for cases with more than one reinfection (Figure 6). The analysis revealed that the average interval between HEV reinfections was 39.7 months (range: 6.1–218.6, SD = 38.1), with a median of 25.2 months. Notably, the infection intervals for cases with initial onset between 2006 and 2014 remained relatively stable, with a median of approximately 30–50 months.

The results of the Cox proportional hazards model (Table 2) indicated that females (HR = 1.306, 95% CI: 1.079–1.582) and individuals aged 41–50 years (HR = 2.341, 95% CI: 1.034–5.301) were at higher risk for HEV reinfection. The Kaplan–Meier (KM) curve (Figure 7) showed that over 80% of reinfection cases had occurred within 100 months after the initial infection.

### 3.2. Spatial Autocorrelation

The results of the global spatial autocorrelation analysis indicated that between 2011 and 2016, as well as in 2018, 2020, and 2022, the HEV reinfection in Zhejiang Province exhibited significant spatial positive autocorrelation (Moran’s I > 0, *p* < 0.05). In the remaining years, the spatial distribution was random (*p* > 0.05), as shown in Table 3.

For these specific years, local spatial autocorrelation analysis was conducted (Figure 8). The results revealed that from 2011 to 2016, there was a noticeable spatial clustering of HEV reinfection cases in Hangzhou. High–high clustering areas were primarily concentrated in central urban areas, including Fuyang District, Yuhang District, Xihu District, Gongshu District, and Shangcheng District. After 2018, the high–high clustering areas shifted to Wenzhou and Taizhou, particularly in Yueqing in Wenzhou and Luqiao and Jiaojiang in Taizhou. Low–low clustering areas were mainly found in Wuyi in Jinhua and Jingning She Autonomous County in Lishui, which were regions with lower reinfection rates of HE.

### 3.3. Spatiotemporal Scan 

From 2005 to 2023, the spatiotemporal scan analysis identified two significant clusters of HEV reinfections in Zhejiang Province, categorized into primary and secondary clusters based on the log-likelihood ratio (LLR) values, as detailed in Table 4. The primary cluster was located in Hangzhou, with the cluster period spanning from 2011 to 2019, and an LLR value of 102.1 (*p* < 0.001). The secondary clusters were identified in Wenzhou and Taizhou, with the cluster period from 2015 to 2023, and an LLR value of 54.3 (*p* < 0.001).

## 4. Discussion

Regarding temporal distribution, the reinfection rate of HEV in Zhejiang Province from 2005 to 2023 fluctuated, showing a gradual increase from 2005 to 2020 followed by a significant decline in 2021. This pattern exhibited a noticeable lag compared to the overall incidence of HE, which aligned with the characteristics of reinfection. Analysis of infection intervals for cases with different initial infection years showed that the interval for HEV reinfection was approximately 3–4 years, consistent with the observed lag. Moreover, the rapid increase in HE incidence in recent years suggested a potential rise in reinfection rates in the coming years, highlighting the need for heightened prevention and control efforts.

From the perspective of the reinfection rate, this study reported a reinfection rate of 2.13%, which significantly differed from the 17% documented in a large cohort study [10]. This discrepancy was likely due to differences in the definitions of reinfection and the study populations between the two studies. First, the large cohort study determined the reinfection based on marker profiles of acute infection. In our study, these markers were simply used to diagnose the current infection, and reinfection was defined as confirmed diagnoses occurring six months or more apart. Some patients with reinfection might be mild or asymptomatic cases and did not seek medical attention, which limited us from obtaining those markers and led to a lower observed reinfection rate. Second, the large cohort study involved a community population, where most patients had mild symptoms and did not seek medical care but were finally included in the study. In contrast, our study included only confirmed patients. Currently, there are no accurate data available regarding the reinfection rate of HEV in the population, highlighting the urgent need for more precise measurement methods.

In terms of demographic distribution, reinfection cases were predominantly among middle-aged and elderly individuals, similar to the general HE patients [5,16]. Additionally, this study found that the proportion of females among patients with reinfection was significantly higher than among those with a single infection, and the age at first infection was significantly lower for patients with reinfection compared to single-infection cases. These findings were consistent with previous research [10]. Regarding occupational distribution, patients with reinfection were primarily farmers, commercial service workers, and retirees. This distribution was likely related to the frequent contact between farmers and livestock, as multiple studies identified pig and poultry farmers as high-risk groups for HE [17]. The high proportion of retirees was associated with population aging [2,18]. However, compared to single-infection cases, the proportion of farmers among patients with reinfection was lower, while the proportions of staff, food industry workers, and commercial service workers were significantly higher. This difference could be attributed to higher-risk behaviors, such as dining out, among these groups. This observation highlighted the need to improve monitoring systems and develop targeted prevention strategies for high-risk populations.

In terms of spatial distribution, global spatial autocorrelation analysis in 2005–2023 indicated that HEV reinfection rates were spatially positively correlated in 2011–2016, 2018, 2020, and 2022, while other years showed random distribution. Local spatial autocorrelation analysis revealed that high–high clusters were primarily concentrated in Hangzhou during the period of 2011–2016, with a subsequent shift to coastal areas such as Wenzhou and Taizhou in later years. This suggested a need to consider the risk of HEV transmission through seafood [19,20,21]. Additionally, these cities were also high-incidence areas for HE, with high population density and mobility, indicating that concentrated populations facilitated HEV transmission. Spatiotemporal scanning identified Hangzhou as the primary cluster area for HEV reinfection from 2011 to 2019, and Wenzhou and Taizhou from 2015 to 2023, consistent with the autocorrelation results.

The strengths of our study included being the first investigation into the epidemiological characteristics and spatiotemporal distribution of patients with HEV reinfection. Furthermore, data collection was conducted using the Chinese Disease Surveillance Information Reporting System, with stringent inclusion and exclusion criteria to ensure data accuracy and relevance. The study also employed various statistical analysis techniques to enhance the reliability and validity of the research. Additionally, including data from the COVID-19 pandemic period provided a broader perspective on HEV reinfection.

However, this study had limitations. Firstly, as a cross-sectional study, it could not establish causal relationships between factors. Secondly, differences in healthcare levels across regions may have contributed to the observed disparities in reinfection rates. Thirdly, while HEV reinfection was widespread, some patients with asymptomatic or mild cases may not have sought medical attention, potentially excluding them from the study. Lastly, although we excluded patients with immune deficiencies, some chronic infection cases may still have been included.

## 5. Conclusions

From 2005 to 2020, the rate of HEV reinfection in Zhejiang Province exhibited an increasing trend, which then gradually declined from 2020 to 2023. Female gender and the age group of 41–50 years were identified as risk factors for HEV reinfection. There was a significant spatiotemporal clustering of HEV reinfection in Zhejiang Province, with the primary clusters located in Hangzhou, Wenzhou, and Taizhou.

## Figures and Tables

**Figure 1 viruses-16-01676-f001:**
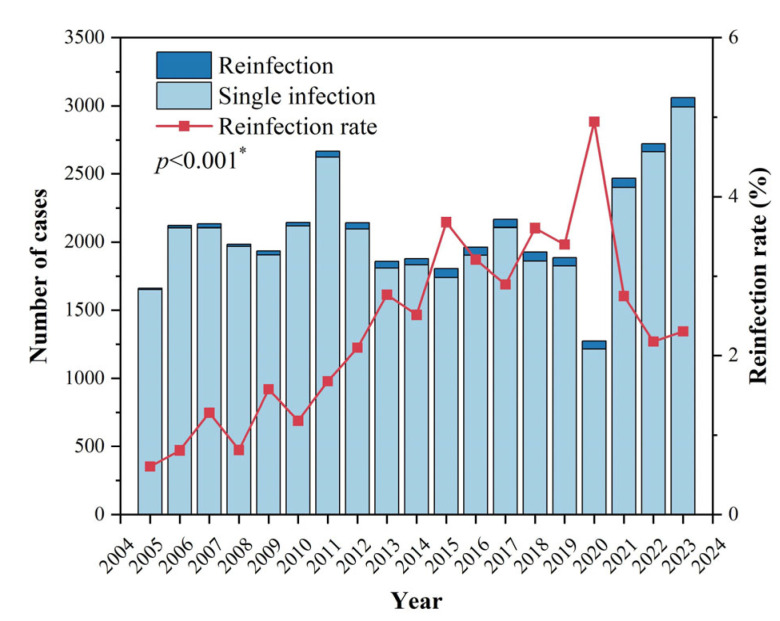
Number of Hepatitis E cases and reinfection rate in Zhejiang Province, 2005–2023. * There were significant differences in annual reinfection rates, which could be divided into three groups. Specifically, the rates in 2012, 2013, 2014, 2021, 2022, and 2023 were significantly higher than those in 2005–2011 and significantly lower than those in 2015–2020.

**Figure 2 viruses-16-01676-f002:**
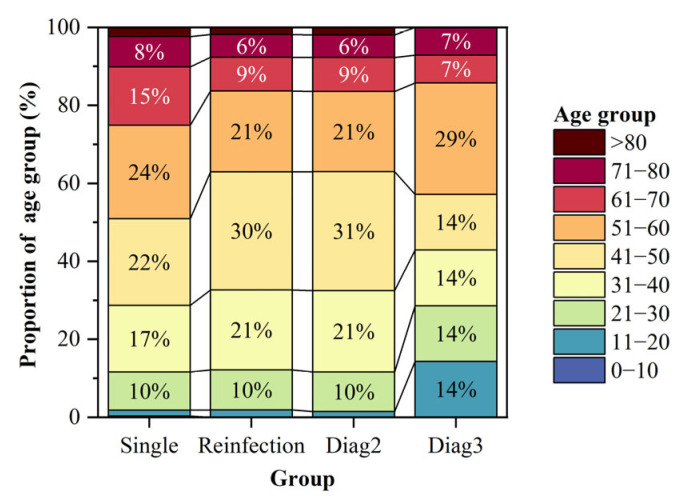
Distribution of age groups at first infection among patients with Hepatitis E with different numbers of infections.

**Figure 3 viruses-16-01676-f003:**
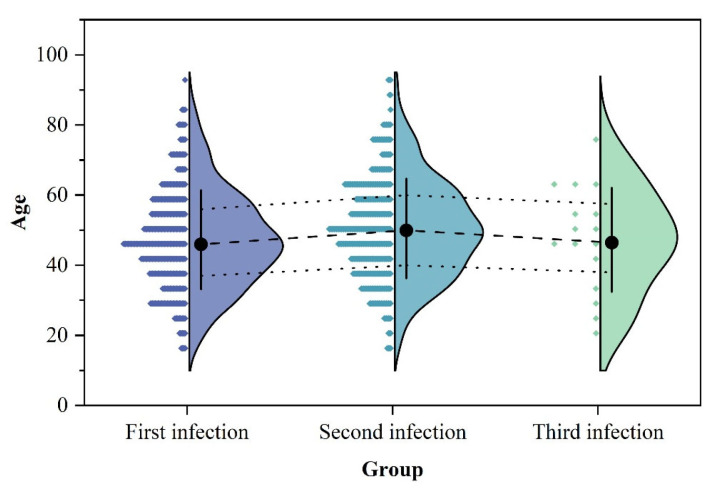
Age distribution of multiple infections among patients with Hepatitis E reinfection in Zhejiang Province.

**Figure 4 viruses-16-01676-f004:**
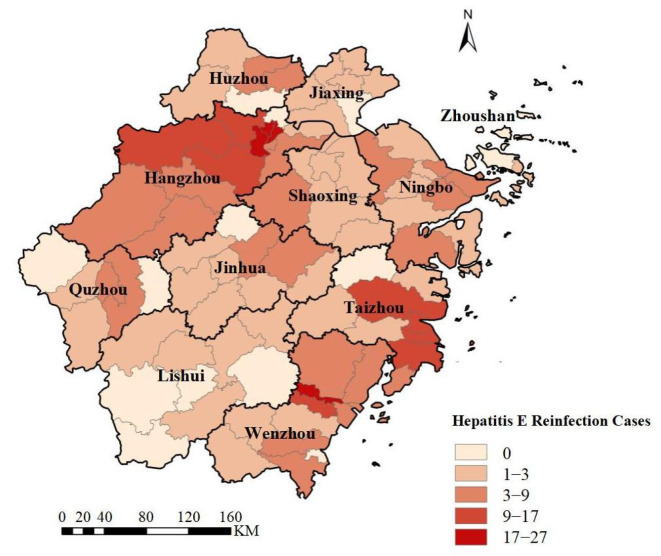
Distribution of Hepatitis E reinfection cases across counties and districts in Zhejiang Province, 2005–2023.

**Figure 5 viruses-16-01676-f005:**
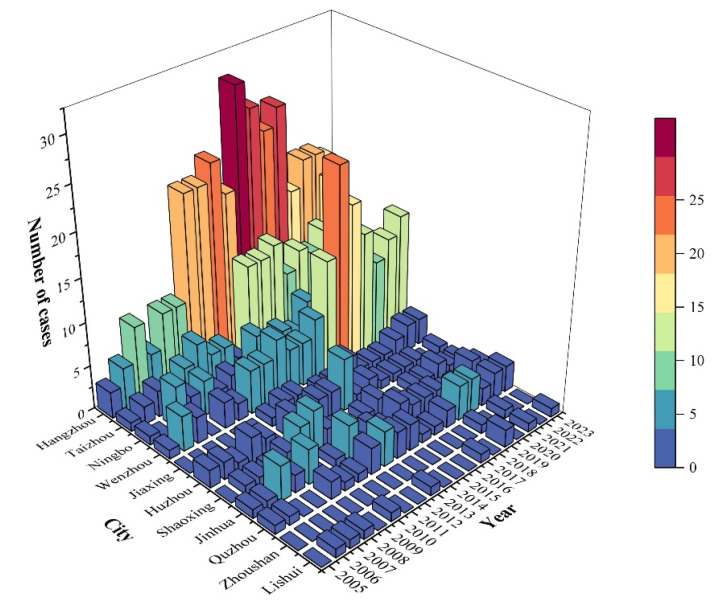
Temporal distribution of Hepatitis E reinfection cases across cities in Zhejiang Province, 2005–2023.

**Figure 6 viruses-16-01676-f006:**
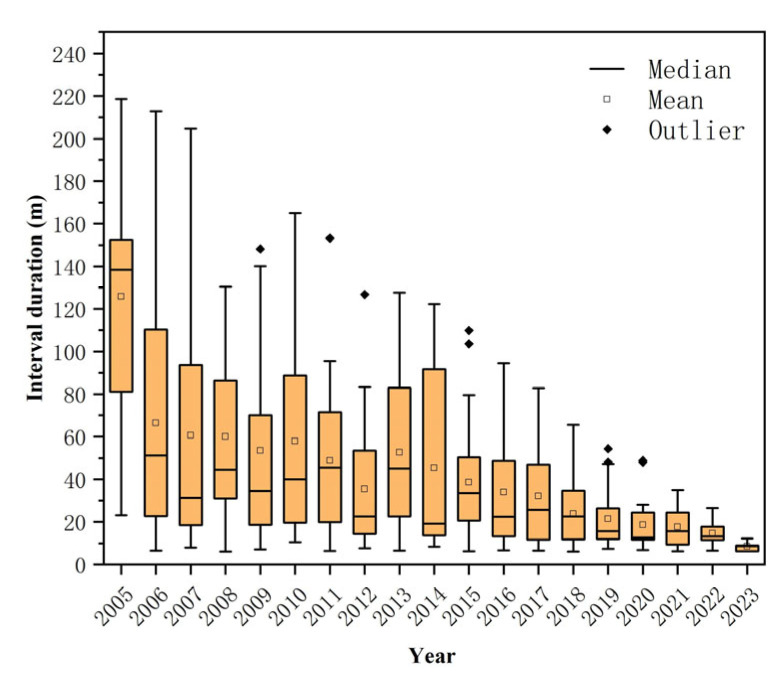
Distribution of infection intervals among cases with different initial onset years.

**Figure 7 viruses-16-01676-f007:**
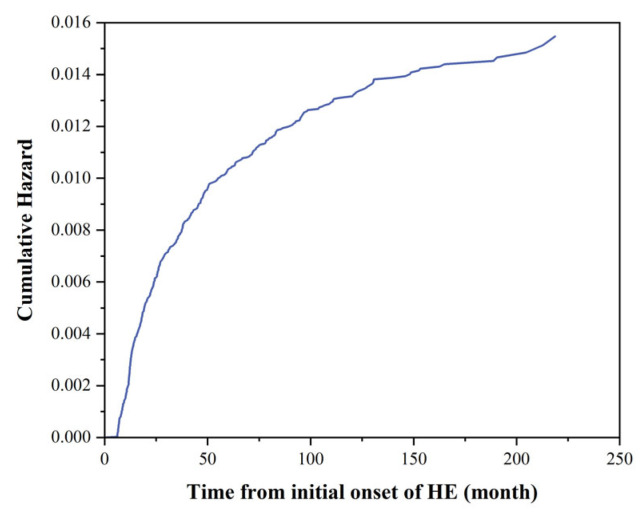
Kaplan–Meier cumulative risk curve of reinfection after initial infection for all patients with Hepatitis E.

**Figure 8 viruses-16-01676-f008:**
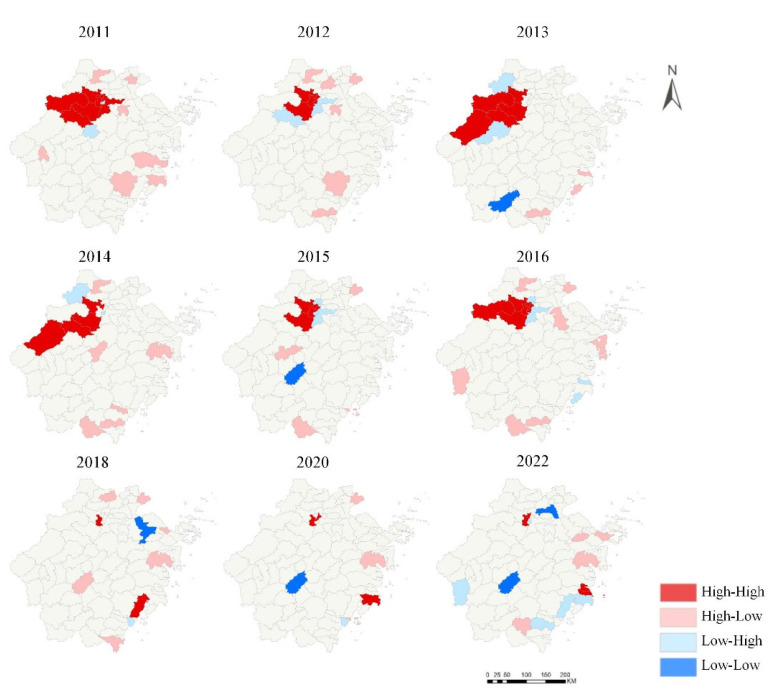
Local spatial autocorrelation analysis of clustering patterns.

**Table 1 viruses-16-01676-t001:** Demographic characteristics of individuals with Hepatitis E in Zhejiang Province.

	Single Infection (n = 38,910)	Reinfection		
		Overall (n = 429)	Infected Twice (n = 413)	Three Times (n = 14)
Gender				
Male	26,145 (67.2)	264 (61.5)	253 (61.3)	9 (64.3)
Female	12,765 (32.8)	165 (38.5)	160 (38.7)	5 (35.7)
Age at first infection	50.0 (39.0, 61.0)	46.0 (37.0, 56.0)	46.0 (37.0, 56.0)	44.0 (28.3, 57.3)
Age at first reinfection	-	50.0 (40.0, 60.0)	50.0 (40.0, 60.5)	47.5 (29.3, 58.8)
Age group				
>0 and ≤10 years	108 (0.3)	0 (0)	0 (0)	0 (0)
>10 and ≤20 years	608 (1.6)	8 (1.9)	6 (1.5)	2 (14.3)
>20 and ≤30 years	3805 (9.8)	44 (10.3)	42 (10.2)	2 (14.3)
>30 and ≤40 years	6630 (17.0)	88 (20.5)	86 (20.8)	2 (14.3)
>40 and ≤50 years	8649 (22.2)	130 (30.3)	126 (30.5)	2 (14.3)
>50 and ≤60 years	9326 (24.0)	89 (20.7)	85 (20.6)	4 (28.6)
>60 and ≤70 years	5840 (15.0)	37 (8.6)	36 (8.7)	1 (7.1)
>70 and ≤80 years	3010 (7.7)	25 (5.8)	24 (5.8)	1 (7.1)
>80 years	934 (2.4)	8 (1.9)	8 (1.9)	0 (0)
City				
Hangzhou	13,487 (34.7)	156 (36.4)	148 (35.8)	8 (57.1)
Ningbo	3635 (9.3)	39 (9.1)	39 (9.4)	0 (0)
Wenzhou	4897 (12.6)	77 (17.9)	72 (17.4)	5 (35.7)
Jiaxing	2297 (5.9)	11 (2.6)	11 (2.7)	0 (0)
Huzhou	1955 (5.0)	16 (3.7)	15 (3.6)	0 (0)
Shaoxing	1924 (4.9)	14 (3.3)	14 (3.4)	0 (0)
Jinhua	2590 (6.7)	19 (4.4)	19 (4.6)	0 (0)
Quzhou	2060 (5.3)	19 (4.4)	18 (4.4)	1 (7.1)
Zhoushan	304 (0.8)	1 (0.2)	1 (0.2)	0 (0)
Taizhou	4417 (11.4)	72 (16.8)	71 (17.2)	0 (0)
Lishui	1344 (3.5)	5 (1.2)	5 (1.2)	0 (0)
Occupation				
Farmers	17,551 (45.1)	150 (35.0)	146 (35.4)	3 (21.4)
Workers in service industry	2850 (7.3)	58 (13.5)	56 (13.6)	2 (14.3)
Manual workers	4725 (12.1)	43 (10.0)	41 (9.9)	2 (14.3)
Persons with unemployment	3174 (8.2)	43 (10.0)	41 (9.9)	2 (14.3)
Retired persons	2970 (7.6)	38 (8.9)	35 (8.5)	3 (21.4)
Office workers	1803 (4.6)	31 (7.2)	30 (7.3)	1 (7.1)
Workers in catering industry	580 (1.5)	17 (4.0)	17 (4.1)	0 (0)
Healthcare workers	218 (0.6)	5 (1.2)	5 (1.2)	0 (0)
Students	379 (1.0)	3 (0.7)	3 (0.7)	0 (0)
School teachers	410 (1.1)	2 (0.5)	2 (0.5)	0 (0)
Unknown	4250 (10.9)	39 (9.1)	37 (9.0)	1 (7.1)
Infection interval	-	770 (389, 1624)	770 (389, 1624)	555 (347, 805.5)

**Table 2 viruses-16-01676-t002:** Multivariate analysis of Hepatitis E reinfection by the Cox proportional hazards model.

	Number of Reinfections	Number of Patients Without Reinfection	Hazard Ratio (95% CI)	*p*-Value
Gender, n (%)				
Male (ref.)	264 (61.5)	26,145 (67.2)	-	-
Female	165 (38.5)	12,765 (32.8)	1.306 (1.079, 1.582)	0.006
Age group				<0.001
>0 and ≤10 years	0 (0)	108 (0.3)	- ^1^	0.916
>10 and ≤20 years	8 (1.9)	608 (1.6)	2.512 (0.929, 6.793)	0.070
>20 and ≤30 years	44 (10.3)	3805 (9.8)	1.616 (0.689, 3.790)	0.270
>30 and ≤40 years	88 (20.5)	6630 (17.0)	1.930 (0.844, 4.413)	0.119
>40 and ≤50 years	130 (30.3)	8649 (22.2)	2.341 (1.034, 5.301)	0.041
>50 and ≤60 years	89 (20.7)	9326 (24.0)	1.536 (0.673, 3.505)	0.308
>60 and ≤70 years	37 (8.6)	5840 (15.0)	1.010 (0.427, 2.385)	0.983
>70 and ≤80 years	25 (5.8)	3010 (7.7)	1.279 (0.527, 3.108)	0.587
>80 years (ref.)	8 (1.9)	934 (2.4)	-	-

^1^ In the age group >0 and ≤10 years, the number of reinfection cases was zero.

**Table 3 viruses-16-01676-t003:** Global spatial autocorrelation analysis of Hepatitis E reinfection rates in Zhejiang Province, 2005–2023.

Year	Moran’s I	Z-Score	*p*-Value
2005	0.001679	0.214633	0.830053
2006	0.014455	0.440556	0.659534
2007	0.074848	1.388639	0.164942
2008	−0.088769	−1.402622	0.16073
2009	0.005209	0.265244	0.790821
2010	−0.021957	−0.011236	0.861306
2011	0.286387	4.68540	0.000003
2012	0.290154	4.812043	0.000001
2013	0.149511	2.915539	0.003551
2014	0.113169	2.023546	0.043017
2015	0.220563	3.598248	0.00032
2016	0.252269	4.103154	0.000041
2017	0.080457	1.729372	0.083743
2018	0.136348	2.395164	0.016613
2019	0.108564	1.849196	0.06443
2020	0.14863	2.497568	0.012505
2021	0.024647	0.597199	0.550375
2022	0.157693	2.592358	0.009532
2023	−0.025675	−0.220475	0.825502

**Table 4 viruses-16-01676-t004:** Spatiotemporal scan analysis of Hepatitis E reinfection cases in Zhejiang Province, 2005–2023.

Cluster Type	Time Frame	Cluster Areas	Observed Cases	Expected Cases	LLR	RR	*p*-Value
Primary	2011–2019	Hangzhou	208	69.25	102.1	3.61	<0.001
Secondary	2015–2023	Wenzhou, Taizhou	232	115.86	54.3	2.36	<0.001

## Data Availability

The datasets used and/or analyzed during the current study are available from the corresponding author upon reasonable request.

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
