# Peer review of "Epidemiological Characteristics and Spatiotemporal Clustering of Symptomatic Hepatitis E Virus Reinfection in Zhejiang Province, 2005–2023"

_viruses, 2024, doi:10.3390/v16111676_

Round 1
Reviewer 1 Report
Comments and Suggestions for Authors
The authors conducted an analysis of data from the Hepatitis E case database from Zhejiang Province, utilizing the Chinese Disease Surveillance Information Reporting System, to investigate the epidemiological characteristics, spatiotemporal clustering, and high-risk populations of symptomatic hepatitis E virus (HEV) reinfection from 2005 to 2023 in Zhejiang Province, China. A total of 41,187 hepatitis E cases were documented, among which 877 reinfection cases (2.13%) were identified, involving 429 individuals. Of these 429 patients, 413 experienced two episodes of HEV infection, 14 experienced three episodes, 1 experienced four episodes, and 1 experienced five episodes.
This study provides valuable epidemiological and clinical insights into HEV reinfection. However, there are several significant concerns that need to be addressed, as outlined below:
Major Comments:
1. The authors mention that the study used the Hepatitis E case database from Zhejiang Province (Lines 73-74). However, it remains unclear how this database can be accessed and whether the "Diagnostic Criteria for Hepatitis E" includes virological testing, such as the detection of IgM anti-HEV antibodies and HEV RNA positivity. Clarification of these diagnostic criteria is essential.
2. The authors suggest that some patients with asymptomatic or mild cases may not have sought medical attention, which could have excluded them from the study (Lines 281-282). If asymptomatic or mild cases were included in the study, the manuscript should explain how these cases were diagnosed and incorporated into the database. If such cases were not included, the title should be revised to specify “Symptomatic” Hepatitis E Virus Reinfection (Line 3).
3. The information on 16 cases with three (n=14), four (n=1), or five (n=1) HEV reinfections is of significant interest to readers. A table presenting demographic characteristics, as well as laboratory and virological data for each reinfection episode, should be provided to facilitate a more detailed understanding of these cases.
4. The reinfection rate of 2.13% reported among symptomatic hepatitis E cases (Line 129) is notably lower than the 17% reported in a large cohort study conducted in rural eastern China (Ref. 10). The authors should explore and discuss potential reasons for this substantial discrepancy.
5. The authors state that Hangzhou, Wenzhou, and Taizhou are high-incidence areas for hepatitis E due to high population density and mobility, which facilitate HEV transmission (Lines 266-268). Supporting data for each of all 11 cities in Zhejiang Province should be provided in a new table, detailing the number of first-time and reinfection cases (second, third infections) per 10,000 population.
Minor Comments:
1. Lines 78-79: The term "des gender" is unclear and the sentence in which it appears is incomplete. This should be revised for clarity.
2. Line 104: The name of the company or a relevant citation should be provided for SPSS 25.0 or R 4.1.3.
3. Line 107: The name of the company or a relevant citation should be provided for ArcGIS 10.7 software.
4. Line 113: The name of the company or a relevant citation should be provided for SATScan 10.1.2 software.
5. Lines 123-125: The sentences in these lines should be deleted as they are unnecessary.
6. References 1 and 13 are incomplete, lacking page numbers. These should be updated to ensure completeness.
7. Table 1 and Tables S1-S3: The occupation categories (e.g., worker, retired, staff, food industry) should be reviewed for consistency and accuracy.
8. Figure 1: It should be clarified in the figure legend whether the differences in annual reinfection rates are statistically significant.
Comments on the Quality of English LanguageMinor editing of English language is required. For example, in Table 1 and Tables S1-S3, the occupation categories (e.g., worker, retired, staff, food industry) should be reviewed for consistency and accuracy.
Reviewer 2 Report
Comments and Suggestions for Authors
The analysis was carried out on a very limited population, therefore the interest could be marginal, it is advisable to extend the research to other populations, possibly even outside China, to carry out a more accurate and extensive study.
Comments on the Quality of English LanguageThe English is accurate and understandable
Round 2
Reviewer 1 Report
Comments and Suggestions for Authors
I have carefully reviewed the revised manuscript and the response letter. I am pleased to note that the manuscript has been revised in accordance with my previous comments. Therefore, I am happy to approve the revision for publication in Viruses.